# Protoplast-Based Transient Expression and Gene Editing in Shrub Willow (*Salix purpurea* L.)

**DOI:** 10.3390/plants11243490

**Published:** 2022-12-13

**Authors:** Brennan Hyden, Guoliang Yuan, Yang Liu, Lawrence B. Smart, Gerald A. Tuskan, Xiaohan Yang

**Affiliations:** 1Biosciences Division, Oak Ridge National Laboratory, Oak Ridge, TN 37830, USA; 2School of Integrative Plant Sciences, Cornell AgriTech, Cornell University, Geneva, NY 14456, USA

**Keywords:** *Salix*, protoplasts, gene editing

## Abstract

Shrub willows (*Salix* section Vetrix) are grown as a bioenergy crop in multiple countries and as ornamentals across the northern hemisphere. To facilitate the breeding and genetic advancement of shrub willow, there is a strong interest in the characterization and functional validation of genes involved in plant growth and biomass production. While protocols for shoot regeneration in tissue culture and production of stably transformed lines have greatly advanced this research in the closely related genus *Populus*, a lack of efficient methods for regeneration and transformation has stymied similar advancements in willow functional genomics. Moreover, transient expression assays in willow have been limited to callus tissue and hairy root systems. Here we report an efficient method for protoplast isolation from *S. purpurea* leaf tissue, along with transient overexpression and CRISPR-Cas9 mediated mutations. This is the first such report of transient gene expression in *Salix* protoplasts as well as the first application of CRISPR technology in this genus. These new capabilities pave the way for future functional genomics studies in this important bioenergy and ornamental crop.

## 1. Introduction

Willow (*Salix* spp.) is a diverse genus containing over three hundred species, with a native range spanning across every continent except Australia and Antarctica, including particularly high genetic diversity in the temperate regions of North America and Eurasia [1]. Willows are used for a variety of human purposes, including bioenergy and biomass production, phytoremediation and ecological restoration, as ornamentals, and in cultural applications including basket weaving and cricket bat production [2]. Biomass production of willows for carbon sequestration and bioenergy is of particular interest due to their perennial nature, low inputs, and ability to grow on marginal land [3]. One of the greatest limitations to the advancement of willow genetics and biology for biomass production is an inability to regenerate *Salix* shoots from tissue culture, preventing the development of stable transgenic lines and hindering further functional genomics studies. Transformation and transient expression have so far been limited to callus tissue and hairy root systems [4,5]. However, both expression systems are technically demanding, requiring at least several weeks for a single experiment as well as the use of tissue culture and antibiotic selection protocols.

Protoplast expression experiments offer a tractable alternative to callus and hairy root expression systems as they can be completed in as little as two days, are obtained in large quantities, and for *Salix* do not require the use of tissue culture. Protoplast transformation, CRISPR/Cas9 knockout, and expression protocols are well established in the closely related genus *Populus* [6]. Protoplast isolation has also been demonstrated in *Salix* [7], although overexpression and CRISPR/Cas9—based gene editing have not been reported. In this study, we show transient overexpression of multiple reporter genes in *Salix* leaf protoplasts and report on the first application of gene editing technology in willow through CRISPR/Cas9-induced mutations. These results represent a substantial advancement in willow functional genomics and biotechnology and establish the ability to perform a variety of future studies to validate the precise molecular mechanisms and function of genes in willows.

## 2. Results and Discussion

Six technical replicates of the *S. purpurea* protoplast extractions were performed, and typical protoplast yields for preparations using six leaves ranged from 8 × 10^5^ to 20 × 10^5^ protoplasts mL^−1^. This represents an improvement over current protoplast isolation methods in *Populus*, which require using leaves grown in tissue culture conditions. We demonstrated that large numbers of protoplasts can be obtained from greenhouse-grown leaves within two weeks of starting cuttings. This, coupled with the speed and ease of propagating willow cuttings under greenhouse conditions, provides an effectively unlimited supply of protoplasts and suggests that there is great potential for the development of high-throughput protoplast applications in this genus.

Transient expression in leaf protoplasts was successful using four vectors expressing different fluorescent proteins (Figure 1A–D). Transformation of protoplasts with constructs containing 35S: sfGFP (Figure 1A), 35S: eGFP (Figure 1B), 35S: eYGFPuv (Figure 1C), and 35S: mCherry (Figure 1D) all produced easily detectable fluorescence signals after 24–48 h of incubation. Overall, we obtained satisfactory transformation efficiencies, with all but one construct showing greater than 45% efficiency (Figure 1E). The observed variation in transformation efficiency between constructs is likely due to many factors, including cell viability and technical error during transformation. Improvement in transformation efficiency can be made through future experimental optimization for *S. purpurea* and other species in the same genus. These results are the first report of successful *Salix* protoplast transformation.

In this study, we demonstrated that CRISPR/Cas9 is functional in willow using our protoplast transient assay platform. Illumina sequencing of PCR amplicons for each target gene revealed mutations at each of six gRNA target sites (Figure 2A,B). Total sequenced reads were 351,532 for the *ANGUSTIFOLIA* amplicon, 434,833 for the *PHYA* amplicon, and 70,575 for the *FRIGIDA* amplicon. Gene-editing efficiencies for each gRNA were 0.44% and 0.40% for *ANGUSTIFOLIA* gRNA1 and gRNA2, respectively, 0.14% and 0.17% for *PHYA*, and 17.27% and 0.54% for *FRIGIDA*. Most of the mutations occurred at 3 bp upstream of PAM sequence. Only small deletions ranging from one to seven bp and single bp substitutions were observed in each individual target site. One high activity gRNA, *FRIGIDA* gRNA1, was identified by our transient assay, suggesting that our system may be used to test the activity of different gRNAs (Figure 2B). Large deletions were also observed between the target regions of two *ANGUSTIFOLIA*-specific gRNAs, indicating that multiplexed gene editing can be achieved in willow using CRISPR/Cas9. However, all but one gRNA had an editing efficiency of below 1%, whereas typical protoplast CRISPR knockout experiments analyzed with next-generation sequencing produce 2% efficiency or greater [8,9]. This low rate of mutation may be due to either poor transformation efficiency or low gRNA activity. In the future, a fluorescent marker can be included in the CRISPR/Cas9 construct as a reporter of transformation efficiency. Refinement of gRNA and PAM sites to optimize for use in *Salix* and testing of different promoters and terminators for Cas9 and gRNAs such as CmYLCB, NOS, or ACTIN could also improve efficiency in the future [10]. Nevertheless, the result obtained from this study provides compelling evidence that the CRISPR/Cas9 system is indeed viable in *Salix* protoplasts.

Taken together, the findings in this study represent a substantial advancement in *Salix* genomics research and can serve as the basis for future functional genomics experiments in this understudied crop. In demonstrating that both transient overexpression and CRIPSR/Cas9-based gene editing are active in *Salix*, we propose that additional experimental approaches involving cell suspension cultures, expression of protein-protein interaction constructs, and RNA-Seq and proteomic analysis following gene perturbation are potentially tractable methods. Such studies will dramatically advance our understanding of the *Salix* gene function associated with the traits related to biomass yield, bioenergy production, and horticultural applications.

## 3. Materials and Methods

### 3.1. Plant Materials and Growth Condition

The plant material for protoplast isolation was obtained from fully expanded young leaves harvested from two-week-old greenwood cuttings of *S. purpurea* “94006” grown in a 16h photoperiod at 22 °C in the greenhouse (Figure 3).

### 3.2. Guide RNA Design and Vector Construction

Two CRISPR gRNAs were designed via CHOPCHOP [11] to target the single-copy genes Sapur.002G150400 (*ANGUSTIFOLIA*), Sapur.013G000400 (*PHYA*), and Sapur.15WG127800 (*FRIGIDA*) based on the *S. purpurea* genome (v5.1) from Phytozome [12]. The two gRNAs targeted sites 100–200 bp apart in the first exon of each gene (Figure 2A). The gRNAs and scaffolds for each target gene were synthesized by Integrated DNA Technologies (Coralville, IA, USA) and separated by tRNA sequences. Each pair of gRNAs was ligated into a pKSE401 vector (Addgene, Plasmid #62202) containing an *Arabidopsis* U3 promoter driving the gRNAs and scaffolds, as well as a SpCas9 nuclease from *Streptococcus pyogenes* driven by a cauliflower mosaic virus 35S promoter [13]. The gene editing constructs were created using PARA [14].

### 3.3. Protoplast Isolation

Protoplast isolation and PEG-mediated transformation protocol described by Yuan et al., 2021 was used to isolate *S. purpurea* protoplasts with the following modifications: leaf digestion was performed in dark conditions at 30 rpm for four hours and following digestion, the enzyme-protoplast solution was immediately diluted without handshaking [15]. The final concentration of *S. purpurea* protoplasts was adjusted to approximately 8 × 10^5^ protoplasts/mL. Protoplasts were transformed with constructs containing each of the following reporter proteins: sfGFP (Addgene, Plasmid #80129), eGFP (Addgene, Plasmid #64401), eYGFPuv [15], and mCherry (modified Addgene Plasmid #64401), each with a hygromycin plant selectable marker and kanamycin bacteria selectable marker. Protoplasts were incubated for 24–48 h prior to imaging. The percentage of cells with fluorescence signal for each fluorescent protein was calculated based on confocal microscope imaging. The images of three independent scopes were taken under a confocal microscope with 10 μL protoplasts per sample. Cell counting in each image was conducted manually.

### 3.4. Analysis of CRISPR/Cas9-Induced Mutation Efficiency

Following transformation with each of the three constructs containing CRISPR gRNAs, protoplasts were incubated for 48 h. Protoplast DNA was extracted according to a protocol by Weigel and Glazebrook, 2009 [16] PCR amplification of each target region was performed using Q5 polymerase (New England Biolabs, Ipswich, MA, USA). Next-generation sequencing of amplicons was performed via Illumina 2 × 150 bp platform (Genewiz, South Plainfield, NJ, USA) and the mutation profile for each gRNA was quantified using Cas-Analyzer [17]. The efficiency of CRISPR/Cas9 mutations was determined by calculating the number of Illumina reads supporting mutation events at each gRNA divided by the total number of sequenced reads.

## Figures and Tables

**Figure 1 plants-11-03490-f001:**
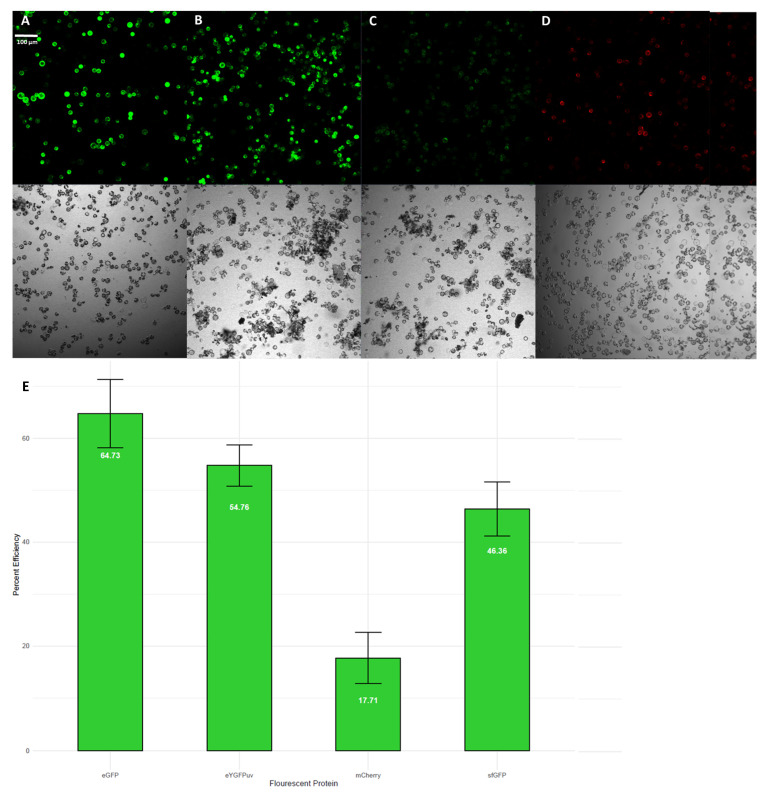
(**A**) sfGFP overexpression; (**B**) eGFP overexpression; (**C**) eYGFPuv overexpression; (**D**) mCherry overexpression; (**E**) percent transformation efficiency of each fluorescent protein tested shown as mean ± standard deviation of three independent scopes.

**Figure 2 plants-11-03490-f002:**
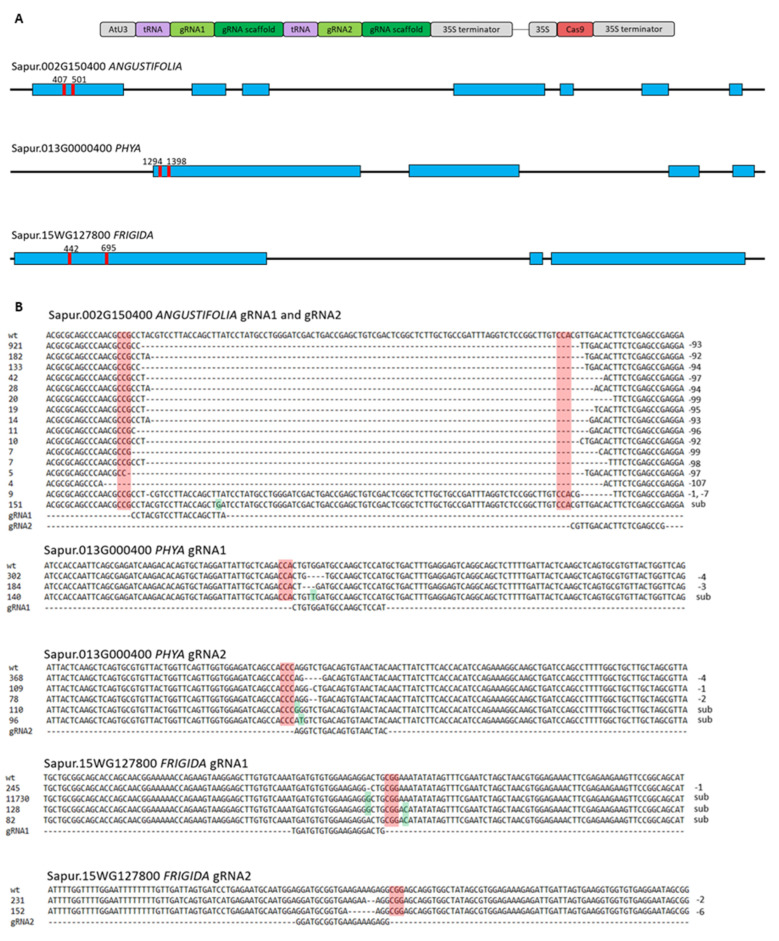
(**A**) Schematic showing the T-DNA design for CRIPSR/Cas9 mediated knockout, exons are shown in blue and gRNA binding sites in red, relative gene lengths are not shown to scale; (**B**) Alignments of CRISPR mutations with read counts (**left**) and mutation type (**right**) for each of the six gRNAs. For brevity, alternative haplotypes with the same mutation were combined, PAM sequences are highlighted in red, and substitutions are highlighted in green.

**Figure 3 plants-11-03490-f003:**
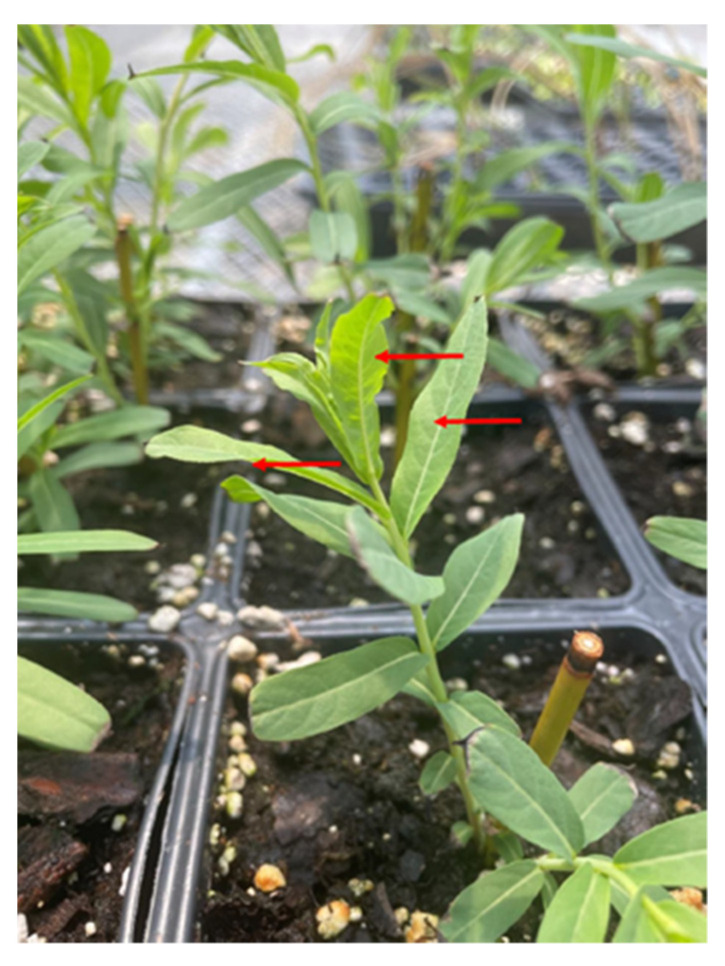
Image of two-week-old cuttings of *Salix purpurea* clone 94006, arrows point to leaves used for protoplast extraction.

## Data Availability

Not applicable.

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
