# Peer review of "Protoplast-Based Transient Expression and Gene Editing in Shrub Willow (Salix purpurea L.)"

_plants, 2022, doi:10.3390/plants11243490_

Round 1

Reviewer 1 Report

In this article, the authors describe a methodology for transfection of protoplasts from Salix sp. with the use of vectors expressing several fluorescent proteins. Then, they demonstrate the ability of using the same transfection protocol to induce the editing, with different efficiencies, of 3 different genes.

The results were well presented and attest that both protoplast transfection and genomic editing were achieved with different degrees of efficiency.

For the framing of the article in the Bried Report class I suggest that the authors merge figure 1 and figure 2 into a single figure. Likewise, figure 5 must precede figure 3 and both compose the same new figure "2".

Author Response

We thank the reviewer for the feedback. As requested, Figures 1 and 2 have been merged to create a new “Figure 1” and Figures 3 and 5 have been merged to create a new “Figure 2”, Figure 4 has been renamed “Figure 3”.

Reviewer 2 Report

In this study, authors present an efficient method of protoplast isolation and transient transformation for Salix purpurea L, as well as the first case of CRISPR/Cas-mediated gene editing of a plant of the Salix genus. The manuscript is well written, the results are clear, and the methods are sound. Therefore, I recommend this manuscript for publication.

If I may suggest some improvements for future studies, I would recommend using FIJI image processing software to count cells semi-automatically instead of manual cell counting. And to improve editing efficiency, authors could try to use different promoters and terminators for Cas9 and gRNA expression instead of 35S (e.g., here is a good review including this topic https://doi.org/10.1016/j.tplants.2021.06.015).

P.S. correct small typos in the text, for example:

line 154: hyromycin -> hygromycin

Author Response

We thank the reviewer for the feedback. Typos in the text have been corrected. In the discussion of future directions, we have included the testing of different promoters and terminators (lines 103-105). We attempted to use FIJI to analyze our images, but the cells were too close together to correctly partition without extensive manual input, and as a result it was much more practical to manually count cells.